# Balancing Objective Function and Search Nodes in MCTS for Constraint Optimization Problems

## Abstract

Constraint Optimization Problems (COPs) present significant challenges in combinatorial problem solving, traditionally tackled using Branch and Bound (B&B) methods that rely on priority queues and iterative branch selection. While recent methods have employed machine learning to enhance branch selection, they often require extensive raw features and large training datasets, which can be a limitation. In this paper, we address this issue by modeling COPs with a minimal set of raw features, and we concentrate on identifying a satisfactory near-optimal or optimal solution within the top $n$ solutions. We propose a novel heuristic neural network algorithm grounded in Monte Carlo Tree Search (MCTS), designed for simultaneous search and training, empowering the neural network to act as an effective heuristic during Backtracking. Furthermore, our approach incorporates encoding COP problems and utilizing graph neural networks to aggregate information about variables and constraints, offering more appropriate variables for assignments. The experiments demonstrate that BalMCTS significantly improves search efficiency, consistently achieving fewer search nodes. For Constraint Satisfaction Problem (CSP) instances, our method reduces the number of search nodes by more than 9.25% compared to state-of-the-art approaches.

## 1 Introduction

Constraint Optimization Problems (COPs) Modi et al. (2005) are fundamental across many domains, including scheduling, routing, and resource allocation Rollon & Larrosa (2014); Bertsekas (2014). These problems are commonly solved by systematic search methods such as Branch-and-Bound (B&B). The efficacy of these solvers depends critically on variable and branch-selection heuristics: effective heuristics reduce explored nodes and enable tractable search, whereas poor heuristics can make otherwise solvable instances infeasible in practice Achterberg & Wunderling (2013); Lodi & Zarpellon (2017). As shown in Figure 1, different heuristics can lead to dramatic variations in the number of explored nodes, making the choice of heuristic a critical factor. This raises a key question: can we design a heuristic that not only performs well on individual problem instances, but also adaptive and efficient in heterogeneous COPs?

Recent research has investigated learning-based heuristics for COPs. Graph neural networks (GNNs) models leverage the structural information of constraint graphs to learn powerful scoring functions, achieving promising results on benchmark instances Wang et al. (2021); Cappart et al. (2023); Gasse et al. (2019). Reinforcement-learning approaches have shown that model-guided decisions can outperform hand-crafted rules on specific benchmarks Parsonson et al. (2023); Cappart et al. (2021). Complementary families of methods—e.g., belief propagation and other message-passing algorithms—have also been adapted to produce fast approximate solutions Deng et al. (2022); Park & Shin (2014). However, existing approaches face practical limitations: many learned policies require extensive offline data and feature engineering, message-passing methods may fail to converge on loopy factor graphs, and purely learned controllers can struggle to trade off solution quality against the cost of exploring large search trees. This suggests that, instead of replacing symbolic search with a learned controller, it is more promising to integrate learning and search.

This paper adopts a hybrid perspective: instead of replacing symbolic search with a learned controller, we integrate a compact, GNN-based heuristic into Monte Carlo Tree Search (MCTS) to jointly search and train. Our method, BalMCTS, is designed with three priciples in mind: (i) use minimal, problem-agnostic features to ensure generality, (ii) leverage the MCTS loop to generate online training data, and (iii) introduce MIROOR, a data-augmentation routine to mitigate sparse rewards. In doing so, BalMCTS learns a policy that balances solution quality with search efficiency.

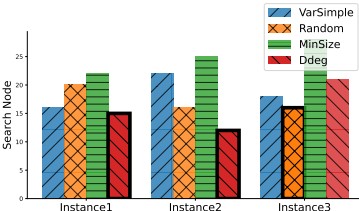

Figure 1: Effect of Variable Selections on Search Node Count Across Diverse COP Instances

We summarize our contributions as follows:

**1. A universal heuristic framework for COP:** We propose a hybrid search-learning method, **Bal**ancing Objective Function and Search Nodes in Monte **C**arlo **T**ree **S**earch (**BalMCTS**), which integrates a neural policy into MCTS, enabling adative and effcient variable selection.

**2. Problem-agnostic feature extraction :**We design a GNN that transforms simple heuristic signals into high-level structural features, supporting generalization across heterogeneous COP families.

**3. MIRROR Operation for Data Augmentation:** We propose the MIRROR, a novel operation that generates diverse variable orderings to alleviate reward sparsity and improve training efficiency.

Our approach integrates MCTS and GNN-based feature extraction to provide a robust, universal framework that adapts to the dynamic nature of COPs, significantly improving search efficiency and solution quality across diverse problem instances.

## 2 PROBLEM FORMULATION

We consider a Constraint Optimization Problem (COP) Deng et al. (2022) defined by the tuple $(X, D, C)$, where $X = \{x_1, \ldots, x_n\}$ is a set of variables, $D = \{d_1, \ldots, d_n\}$ lists finite domains, with $x_i \in d_i$ for each variable, and $C = \{c_1, \ldots, c_m\}$ is a set of constraints. Each constraint $c_j$ has a scope $\mathrm{scp}(c_j) \subseteq X$ and a cost function $f_j$ that maps assignments on $\mathrm{scp}(c_j)$ to a nonnegative cost.

An assignment to all variables is denoted by $\tau = (\tau_1, \ldots, \tau_n) \in \bigotimes_{i=1}^{n} d_i$. The total cost of $\tau$ is

$$F(\tau) = \sum_{c_j \in C} f_j\big(\tau \mid_{\mathrm{scp}(c_j)}\big),$$

where $f_j(\tau|\mathrm{scp}(c_j))$ indicates the cost of $c_j$ with its variables, i.e., $\mathrm{scp}(c_j)$, assigned with values specified by $\tau$.

The goal in a Constraint Optimization Problem (COP) is to find a complete assignment of variables that minimizes the total cost:

$$\tau^* = \operatorname*{arg\,min}_{\tau \in \bigotimes_{i=1}^{n} d_i} F(\tau).$$

A partial assignment $\sigma$ specifies values for a subset $S \subseteq X$ of variables, i.e., $\sigma : S \to \bigcup_{i \in S} d_i$. Applying $\sigma$ to the COP induces reduced domains and may trigger constraint propagation, yielding a reduced problem state $s(\sigma)$. In our framework, each search node corresponds to such a state, and available actions select an unassigned variable and assign it a value, potentially followed by local propagation.

To enable neural encoding, each COP instance is modeled as a bipartite constraint graph: variable nodes represent decision variables, and constraint nodes represent constraints, with edges

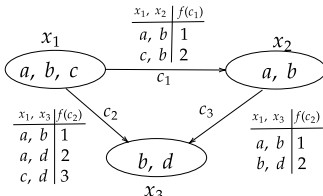

Figure 2: An example of a graph for COP instance (state)

indicating variable participation in constraints. Each node is annotated with compact, problem-agnostic features—such as domain size, number of assigned neighbors, and dynamic constraint tightness—that capture essential local structure. These features are input to a graph neural network (GNN), which computes per-variable embeddings that guide the MCTS policy. *An example COP problem representing as a graph Yokoo et al. (1998) is shown in Figure 2, where the set of variables is $X = \{x_1, x_2, x_3\}$, the set of domains is $D = \{\{a, b, c\}, \{a, b\}, \{b, d\}\}$, the set of constraints is $C = \{c_1, c_2, c_3\}$, where $scp(c_1) = (x_1, x_2)$, $scp(c_2) = (x_1, x_3)$, $scp(c_3) = (x_2, x_3)$, and $f_1(x_1 = a, x_2 = b) = 1$, $f_1(x_1 = c, x_2 = b) = 2$ and so on. The solution for the COP problem is $\tau = (x_1 = a, x_2 = b, x_3 = d)$.*

## 3 OUR BALMCTS APPROACH

In this section, we introduce the `BalMCTS` framework, which unifies graph-based neural representation learning with search-based optimization for constraint problems. The method is organized into three key components: First, we rigorously define the search space by specifying the state and action representations, modeling each problem instance as a structured bipartite graph that encodes variable assignments and constraint relationships. Second, we employ a graph neural network (GNN) to embed these states, enabling the model to capture complex structural dependencies and propagate information between variables and constraints. Third, we integrate these learned representations into a Monte Carlo Tree Search (MCTS) algorithm, where the neural policy guides variable selection and assignment at each search step. This integration allows the policy to balance exploration and exploitation, leveraging both learned structure and search dynamics. The complete workflow of `BalMCTS` is summarized in Figure 3.

### 3.1 GNN-BASED REPRESENTATION

We first provide the definitions of *State* and Action, as shown below.

***Definition 1 (State)*** *We define the state as a graph structure representing a Constraint Optimization Problem (COP) instance, denoted as the COP instance graph. This graph consists of variables and constraints nodes. Notably, the neighbours of each constraint node are variable nodes, and the neighbors of each variable node are constraint nodes. Specifically, the state is an instance for COP defined as $s = \{x_1, x_2, ..., x_n; d_1, d_2, ..., d_n; c_1, ..., c_j\}$, where $x_i$ represents the $i$-th variable node, $d_i$ represents the feasible domain for $x_i$, and $c_j$ represents the $j$-th constraint node.*

***Definition 2 (Action)*** *We define $a$ as operations that select a variable. By assigning a value to the selected variable, the information of nodes in the COP instance graph is modified, leading to a transition from the current state $s$ to a new state $s'$. This state transition is based on operations such as constraint propagation performed on the selected variable. The set of actions is defined as $A(s) = \{a_1, a_2, ..., a_k\} \subseteq \{a_1, a_2, ..., a_n\}$, where $k$ is the number of variables that are not yet assigned values in the state $s$ and $a_i$ represents the action of selecting the $i$-th unassigned variable.*

Having defined the search states and actions, we next describe how to represent them in a way that captures structural dependencies. Specifically, a COP instance can be represented as a constraint graph, as shown in Figure 3. In this paper, we adopt the embedding method proposed by Xu et al. (2019) and Song et al. (2022), utilizing the GNN as a powerful framework for learning expressive vector representations to aggregate information from neighboring nodes. In this GNN variant, $h_x^{(k)}$ and $h_c^{(k)}$ represent the $p$-dimensional embeddings of variable nodes $x$ and constraint nodes $c$ in the $k$-th layer, respectively. The neighboring nodes for $x$ and $c$ are denoted by $\mathcal{N}(x)$ and $\mathcal{N}(c)$.

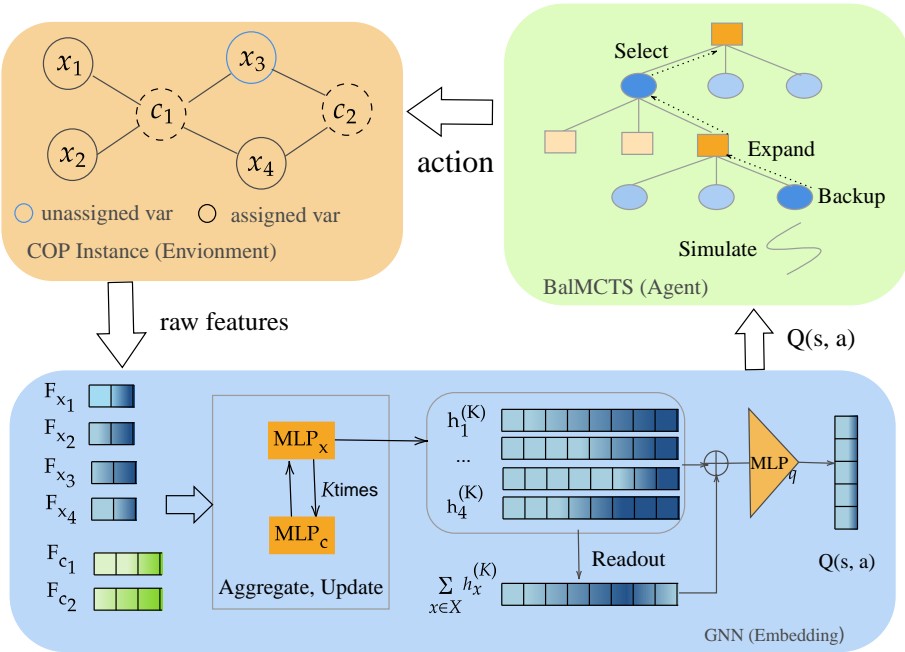

Figure 3: Overview of our `BalMCTS` approach, which integrates Monte Carlo Tree Search (MCTS) with a Graph Neural Network (GNN) to solve Constraint Optimization Problems (COPs). The approach begins with representing the COP instance as a graph, where raw features of variables and constraints are extracted. These features are then processed by the GNN to compute $Q(s, a)$ values, which guide the MCTS agent to iteratively select, simulate, and update actions, with the goal of finding optimal solutions. The selected action is then applied to update the problem state, ultimately finding a solution.

The raw features $F_x$ (with dimension $p_x = 2$) for variable nodes include the 1) current domain size and 2) assigned status, while $F_c$ (with dimension $p_c = 3$) for constraint nodes include the 1) number of bound variables, 2) the dynamic tightness Li et al. (2016), and 3) the minimum cost of the constraints. More specifically, the detailed steps of iterative aggregation and updating of node features for the GNN variant are shown below:

$$h_x^{(0)} = F_x w_x, \ \ h_c^{(0)} = F_c w_c \tag{1}$$

$$h_x^{(k)} = \text{MLP}_x \left( \sum_{c \in \mathcal{N}(x)} h_c^{(k)} : h_x^{(k-1)} : F_x \right) \tag{2}$$

$$h_c^{(k)} = \text{MLP}_c \left( \sum_{x \in \mathcal{N}(c)} h_x^{(k-1)} : h_c^{(k-1)} : F_c \right) \tag{3}$$

Here, the notation $(- : -)$ denotes the concatenation operator. Therefore, the input and output dimensions of $\text{MLP}_x$ are $2p + p_x$ and $p$ respectively, while those of $\text{MLP}_c$ are $2p + p_c$ and $p$. The Eq.(2) and Eq.(3) employ MLP to iteratively aggregate information and update node features for variable node $x$ and constraint node $c$ at $k - 1$ layer, respectively. Eq.(1) is utilized for parameter initialization, where $w_x \in \mathbb{R}^{p_x \times p}$ and $w_c \in \mathbb{R}^{p_c \times p}$ are learnable parameters.

Through K iterations of aggregation and updating, we obtain the variables embedding $h_x^{(K)}$ and constraints embedding $h_c^{(K)}$. To compute $Q(s, a)$, the state $s$ is represented through graph-level pooling, achieved by summing the variable embeddings after $K$ iterations. This pooled representation is then concatenated with the action $a$ and fed into $\text{MLP}_q$ to obtain $Q(s, a)$:

$$Q(s, a) = \text{MLP}_q \left( \sum_{x \in X} h_x^{(K)} : h_a^{(K)} \right) \tag{4}$$

where the input and output dimensions of $\mathrm{MLP_q}$ are $2p$ and 1 respectively. $h_a^{(K)}$ is simply the K-layer embedding $h_x^{(K)}$ of that variable.

## 3.2 MONTE CARLO TREE SEARCH FORMULATION

We adapt Monte Carlo Tree Search (MCTS) to the setting of constraint optimization, where the search space naturally decomposes into *variable nodes* and *value nodes*. As illustrated in Figure 4, circular nodes correspond to variables, while square nodes represent candidate assignments from their domains. Each root-to-leaf path encodes a partial solution, and the search proceeds through five stages.

**SELECTION:** We choose a $Q(s, a)$ with the minimum score based on a variant of the UCT formulation until reaching a leaf node (variable node). Specifically, When selecting a node in the MCTS, the selection is based on a variant of the UCT formula that combines the Q-value of the node, denoted as $Q(s, a)$, with an exploration term $U(s, a)$. The formula is given as:

$$\pi(s) = \arg\min_{a \in A(s)} (Q(s, a) - c_1 \times U(s, a)) \tag{5}$$

where $\arg\min_{a \in A(s)}(Q(s, a))$ represents select a child of minimum Q-value. The $U(s, a)$ is defined by the UCT exploration formula:

$$U(s, a) = \sqrt{\frac{log(\sum_a N(s, a))}{N(s', a')}} \tag{6}$$

where $N(s, a)$ represents the number of times action $a$ was taken from state $s$ during exploration. $s'$ is a child node of $s$, and $c_1$ parameter denotes the degree of exploration. As the number of iterations increases, the value of $U(s, a)$ decreases, resulting in more accurate results.

**EXPANSION:** Due to the presence of the multiple desirable values associated with leaf nodes, one value node is randomly selected for expansion, generating all child nodes (variable nodes) associated with the chosen value node. When a variable node $x_i$ is selected by SELECTION, and $x_i$ has a corresponding domain $d_i$, a random assignment is made to $x_i$ to obtain a value node $\{(x_i = j); j \in d_i\}$. Subsequently, all child nodes of variable node $x_i$ are expanded based on the value nodes $(x_i = j)$. The expansion of nodes is conditioned on the set of variables that are yet to be assigned values.

**SIMULATION:** The simulation process involves the random selection of unbound variables for value assignment, and repeating until all variables are assigned values or until constraint propagation reveals the impossibility of finding a solution. The termination states are defined as either identifying a solution or conclusively establishing the absence of a solution. Also, the number of simulation assignments is at most equal to the greater depth of the tree, i.e. the number of variables.

**MIRROR:** Due to the sparsity of solutions during training, data augmentation is essential to obtain

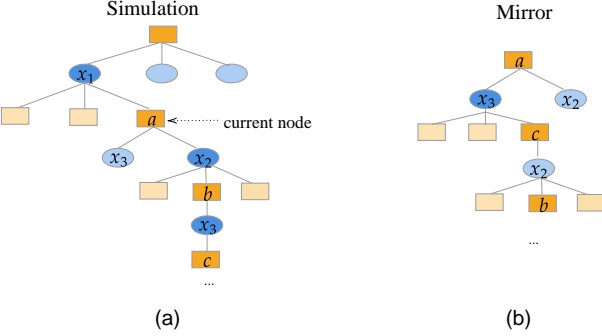

Figure 4: `BalMCTS`. Yellow nodes represent value nodes, and blue nodes represent variable nodes.

sufficient training data. We propose two strategies: (1) discovering new solutions, and (2) generating diverse search paths from the same solution by varying variable selection orders. The MIRROR stage achieves both objectives through a solution augmentation process. We recognize an important fact: **rearranging the order of variables can result in the same solution, but generates different**

**search nodes.** For example, consider a solution $(x_1 = a, x_2 = b, x_3 = c)$ as shown in Figure 4(a). Assigning variables in the order $x_1 \rightarrow x_2 \rightarrow x_3$ yields $(x_1 = a, x_2 = b, x_3 = c)$; alternatively, the order $x_1 \rightarrow x_3 \rightarrow x_2$ may yields the same solution through a different search path in the MCTS tree. The MIRROR stage leverages this by fixing partial solutions and exploring alternative variable orders to produce diverse training data. When a solution is simulated from current node, the node undergoes the MIRROR operation. Specifically, MCTS is conducted with the current node as the root node as shown in Figure 4 (b). Note that this MIRROR operation is independent to the original root node, and its purpose is solely to enhance data diversity.

**BACKUP:** If the current node's objective function is lower than the historical best, $\tau(s)$ is not updated unless the new value is better. During the BACKUP phase, two key values are propagated: the simulation outcome (success = 1, failure = 0) and the solution, which updates $\tau(s)$ from the leaf to the root. For instance, node $x_1$ with a simulation outcome of 3 successes out of 5 ($r(s) = 3/5$) chooses the lowest objective value from its children ($\tau(x_1) = \min(\tau(x_1 = j | j \in d_1))$). This process is replicated for other nodes like $x_2$. Each subtree must maintain its distinct objective value, adjusting only when a simulated solution has a better objective. On top of this, we calculate the target value $y$ using $\tau(s)$ and $r(s)$ through Eq.7 and choose the minimum $y$ for backpropagation.

### 3.3 Training algorithm

Our training algorithm `BalMCTS` is based on MCTS and Double Deep Q-Network (DDQN). DDQN maintains two networks, namely *online network* $Q_w$ and *target network* $Q_{\bar{w}}$. Specifically, $Q_{\bar{w}}$ is a periodical copy of $Q_w$. In each state of the Selection phase, it utilizes the prediction value $Q_{\bar{w}}(s, a)$ for selecting actions among child nodes of state $s$, and adds tuple $(s, a, Q_{\bar{w}}(s, a))$ to experience buffer $\mathcal{R}$ with size $\mathcal{M}$. Then a mini-batch of $\mathcal{B}$ transitions is sample from $\mathcal{R}$ to updat the parameters of the online network $Q_w$ by performing gradient descent to minimize the squared loss between $Q_w(s, a)$ and the following target:

$$y = c_3 \times g(\tau(s')) + c_4 \times (1 - r(s')) \tag{7}$$

where $s'$ represents the state obtained by choosing action $a$ in the state s, denoted as $s' \leftarrow (s, a)$, and $g(\tau(s'))$ denotes the gap of $\tau(s')$. The detailed algorithm is presented in the 1.

Algorithm 1 summarizes the procedure: for each training instance $p \in \mathcal{P}$, MCTS iteratively performs SELECTION, EXPANSION, SIMULATION, and BACKUP up to $\mathcal{T}_{\max}$. Candidate assignments are generated via simulations; feasible ones trigger a MIRROR update, while all outcomes propagate rewards and gaps along the path. Experience tuples are then added to $\mathcal{R}$, and $Q_w$ is updated accordingly. After processing each instance, $\bar{w}$ is synchronized with $w$.

### 3.4 Prompting Large Language Models for Variable Selection in Constraint Optimization Problems

As the capabilities of large language models advance, their reasoning abilities are becoming increasingly powerful, capable of solving complex reasoning and planning tasks. Prompt engineering is crucial for guiding the reasoning outcomes of these large models. Hence, we add additional experiments designing prompt techniques to guide GPT in variable selection for constraint optimization problems (COPs). Due to token length and cost limitations, we need to extract features that can effectively describe the COP problem. The selected features are as follows: (1) Number of variables, (2) Domain length of each variable, (3) Number of constraints on each variable, (4) Mean value each variable can take and (5) Partial assignment. The prompt design strategy and concrete interaction examples are detailed in Appendix A. The usage of large language models in constraint optimization problems is detailed in Appendix B.

### 3.5 Theoretical Analysis

We give two compact results that justify (i) why partial-assignment augmentation such as MIRROR increases the chance of finding feasible solutions in sparse COPs, and (ii) why increasing the training sample size reduces the learned policy's generalization error.

**Theorem 1** *Let $n$ be the total number of variables, $d$ be the domain size of each variable, $m$ be the number of fixed variables at the current node, $|S|$ be the total number of solutions, and $|S_{fixed}|$ be*

---

**Algorithm 1** Balancing Objective Function and Search Nodes in Monte Carlo Tree Search (`BalMCTS`)

---

**Input**: COP instances $\mathcal{P}$
**Parameter**: $\mathcal{T}_{\max}, \mathcal{N}$
**Output**: $\bar{w}$

 1: Initialize the experience replay to capacity $\mathcal{M}$
 2: create root node $x_0$ with state
 3: **for** instance $p \in \mathcal{P}$ **do**
 4:    $\mathcal{T} = 0$
 5:    **while** $\mathcal{T} \leq \mathcal{T}_{\max}$ **do**
 6:       a leaf node $x_i \leftarrow \text{SELECTION}(x_0)$
 7:       observe state $s$
 8:       $\Gamma(s) = \text{EXPANSION}(s)$
 9:       **while** $n \leq \mathcal{N}$ **do**
10:          $X' = \text{SIMULATION}(s' \sim \Gamma(s))$
11:          **if** $X'$ is a feasible solution **then**
12:             $\text{MIRROR}(X')$
13:          **end if**
14:          $\tau(s), r(s) = \text{BACKUP}(X')$
15:       **end while**
16:       y = Eq. (7)
17:       Store $(s, a, y)$ in $\mathcal{R}$
18:       Randomly sample a minibatch $\mathcal{B}$ from $\mathcal{R}$
19:       Perform a gradient descent step to update $w$
20:       $\mathcal{T} \leftarrow \mathcal{T} + 1$
21:    **end while**
22:    For every $p$, set $\bar{w} = w$
23: **end for**
24: **return** $\bar{w}$

---

*the number of solutions consistent with the fixed partial assignment. The probability of finding a solution at the current node, $P_{current}$, exceeds the probability at the root node, $P_{root}$, if $|S| < d^m$.*

Detailed proofs are provided in Appendix C.1. This theorem shows that fixing partial assignments in COPs increases the probability of finding a solution compared to searching from the root node, particularly in sparse problems. This justifies the MIRROR operation in `BalMCTS`, which improved training data in sparse COPs by operating diverse search paths.

**Theorem 2** *Let $\mathcal{X}$ be the input space, $\mathcal{Y} = [0, 1]^n$ be the continuous output space, where $n$ is the dimension of the output vector. A neural network $f_\theta : \mathcal{X} \rightarrow \mathcal{Y}$, where is trained on a dataset $D = (x_i, y_i)$, where $x_i \in \mathcal{X}, y_i \in \mathcal{Y}$, and $N$ is the sample size, with samples drawn from a joint distribution $p(x, y)$. When increasing sample size $N$ the generalization error $\epsilon_g$ will decrease.*

Detailed proofs are provided in Appendix C.2. The second theorem shows that larger training samples reduce generalization error, improving neural network performance on unseen data. This supports the `BalMCTS` training strategy, confirming that MIRROR-generated training data improve generalization, critical for robust sparse COP solving.

The theorems collectively validate the efficacy of MIRROR operation in the `BalMCTS` framework.

## 4 EXPERIMENTS

We evaluate `BalMCTS` on three instance-generated tasks to assess its efficiency and generalization. Our focus is on optimizing the number of explored nodes. Experimental setup, baselines, and implementation details are given in Appendix D.1.

## 4.1 EVALUATION METRICS

We assess performance using four complementary metrics. **Search Node** indicates the total number of nodes explored in the search tree before a feasible solution is found. Lower values indicate greater search efficiency and more effective variable ordering. **Gap** indicates the relative difference between the cost of the best solution found and the true optimum, reported as a percentage. A gap of $0.00\%$ indicates that the method found an optimal solution. **Time** indicates the wall-clock time (in seconds) required to reach the first feasible solution. This measures practical runtime efficiency, including both search and inference overheads. **Node Efficiency (NE)** is defined as $\text{NE} = (1 - \text{Gap})/(\text{Search Node})$, which quantifies how much optimality gap is closed per explored node, capturing the effectiveness of each search step in progressing toward optimality. These metrics together provide a comprehensive view of both the quality and efficiency of the search process.

Table 1: The performance of random COPs ($\Delta = 5$) and WGCP ($\Delta = 1$).

| Methods | Random COPs $\langle m = 2, \gamma = 0.7, \beta = 3, \rho = 0.21 \rangle$ | | | | WGCP $\langle m = 2, \gamma = 0.7, \beta = 3, \rho = 0.21 \rangle$ | | | |
|---|---|---|---|---|---|---|---|---|
| | Gap | Time | Search Node | NE | Gap | Time | Search Node | NE |
| | | | | n = 20 | | | | |
| **BalMCTS-GPT** | 0.00% | 4.08s | **1307** | **7.6** $\times 10^{-4}$ | 0.00% | 5.38s | **2415** | **4.1** $\times 10^{-4}$ |
| BalMCTS | 0.00% | 1.52s | 1485 | 6.7 $\times 10^{-4}$ | **0.00%** | 2.44s | 2509 | 3.9 $\times 10^{-4}$ |
| Toulbar2 | 0.00% | **1.11s** | 5891 | 1.7 $\times 10^{-4}$ | 0.00% | 1.11s | 10294 | 9.7 $\times 10^{-5}$ |
| DABP | 2.94% | 4m27s | - | - | 0.00% | 4.62s | - | - |
| OR-Tools | 0.00% | 1.24 | 4291 | 2.3 $\times 10^{-4}$ | 0.00% | **1.09** | 7639 | 1.3 $\times 10^{-4}$ |
| | | | | n = 30 | | | | |
| **BalMCTS-GPT** | 0.00% | 28.44s | **32629** | **3.06** $\times 10^{-5}$ | 0.00% | 46.52s | **65258** | **1.53** $\times 10^{-5}$ |
| BalMCTS | 0.00% | 25.95s | 32927 | 3.03 $\times 10^{-5}$ | **0.00%** | 48.61s | 72731 | 1.37 $\times 10^{-5}$ |
| Toulbar2 | 0.00% | **12.39s** | 124081 | 8.06 $\times 10^{-6}$ | **0.00%** | **31.49s** | 313942 | 3.18 $\times 10^{-6}$ |
| DABP | 14.21% | 7m19s | - | - | 0.00% | 1m46s | - | - |
| OR-Tools | 0.00% | 13.01 | 138219 | 7.23 $\times 10^{-6}$ | 0.00% | 34.81 | 347819 | 2.88 $\times 10^{-6}$ |
| | | | | n = 40 | | | | |
| **BalMCTS-GPT** | 0.00% | 6m19s | 834291 | 1.19 $\times 10^{-6}$ | 0.00% | 9m14s | **95872** | **1.04** $\times 10^{-6}$ |
| **BalMCTS** | 0.00% | 6m25s | **791151** | **1.26** $\times 10^{-6}$ | 0.00% | 9m27s | 98372 | 1.01 $\times 10^{-6}$ |
| Toulbar2 | 0.00% | **4m5s** | 1359620 | 7.35 $\times 10^{-7}$ | **0.00%** | **5m46s** | 1671902 | 5.98 $\times 10^{-7}$ |
| DABP | 23.62% | 12m32s | - | - | 12.62% | 12m43s | - | - |
| OR-Tools | 0.00% | 4m37 | 1400121 | 7.14 $\times 10^{-6}$ | 0.00% | 6m19 | 1827493 | 5.47 $\times 10^{-7}$ |

## 4.2 RANDOM COPs

To rigorously assess generalization, we evaluate our approach on randomly generated COP instances spanning a range of sizes. This scenario is especially demanding due to the diversity and unpredictability of problem structures, where fixed-rule heuristics often underperform. As summarized in Table 1, BalMCTS achieves a 4.9× reduction in search nodes compared to Toulbar2 for $n = 20$, while consistently finding optimal solutions (Gap = 0) with minimal runtime overhead. For larger instances ($n = 40$), both BalMCTS and BalMCTS-GPT demonstrate superior scalability: their node counts grow much more slowly than those of Toulbar2 and OR-Tools, which exhibit steep increases as problem size rises. DABP, as a fully end-to-end method, does not report node counts, but is nonetheless surpassed by BalMCTS in both runtime and solution quality. Notably, BalMCTS-GPT attains the highest node efficiency (NE), indicating that each search step is highly effective in closing the optimality gap; the integration of language model guidance further enhances both node efficiency and robustness across heterogeneous instances.

These gains stem from two key innovations. First, our GNN-based state encoder captures rich structural information about variables and constraints, enabling the learned policy to make context-aware decisions that outperform classical heuristics like Dom/Ddeg. Second, incorporating MCTS into training provides long-horizon credit assignment, allowing the policy to discover search orderings that judiciously balance constraint propagation with solution depth. Unlike purely end-to-end solvers such as DABP, our search-based framework also exploits partial solutions and backtracking, ensuring that each explored node contributes maximally to reducing the optimality gap.

Table 2: Test CSPs results.

| Random CSPs $\langle m = 2, \gamma = 0.7, \beta = 3, \rho = 0.21, \Delta = 0 \rangle$ | | | | | | |
|---|---|---|---|---|---|---|
| Problem Size | **BalMCTS-GPT** | BalMCTS | DRL | Dom/Tdeg | Dom/Ddeg | MinDom | Impact |
| n=15 | **18.04** | 21.97 | 22.21 | 22.81 | 23.05 | 33.57 | 272.51 |
| n=25 | **243.61** | 273.86 | 291.30 | 320.19 | 347.78 | 799.54 | 69885.05 |
| n=30 (n=25) | **1092.67** | 1149.52 | 1237.93 | 1350.92 | 1523.92 | 4160.78 | 318862.66 |
| n=40 (n=25) | **21491.06** | 21547.30 | 23684.04 | 26861.89 | 31807.81 | 136405.52 | 491872.80 |

## 4.3 RANDOM CSPs.

To isolate the impact of variable ordering, we convert the original COPs to CSPs by setting all constraint costs to zero, thereby removing objective guidance and focusing solely on feasibility. This creates a stringent test of our method's ability to efficiently explore the search space without cost-based heuristics.

**Performance on Random CSP Benchmarks.** We benchmark `BalMCTS` and `BalMCTS-GPT` against classical heuristics (Dom/Ddeg, MinDom, Impact) and a DRL baseline on randomly generated CSPs. In this cost-free setting, node efficiency becomes paramount, as all methods must rely exclusively on structural reasoning. Table 2 shows that both `BalMCTS` variants consistently surpass traditional heuristics and DRL across all instance sizes. While the performance gap is modest at $n = 15$, by $n = 25$ our approach already achieves an order-of-magnitude reduction in search nodes, and this advantage grows with increasing problem size.

These CSP results underscore a key strength of our approach: in the absence of objective costs, search efficiency is determined entirely by the quality of variable ordering. Our GNN-based encoder automatically extracts structural features such as constraint tightness and variable centrality, enabling more adaptive and effective branching than fixed-rule heuristics. Moreover, training on a diverse set of instances fosters generalizable strategies that maintain strong performance as problem scale increases.

**Generalization to larger problem instances** To rigorously test scalability, we trained our models on CSPs with $n = 25$ variables and evaluated them on significantly larger instances ($n = 30$ and $n = 40$; see Table 2). Our approach continued to deliver dramatic reductions in search nodes relative to classical heuristics, even under this substantial distribution shift. This sustained performance indicates that the learned policy internalizes structural patterns—such as variable centrality and constraint density—that generalize well beyond the training regime. In contrast, fixed-rule heuristics deteriorate rapidly as problem size increases, unable to adapt to new structural complexities. The consistently strong results of `BalMCTS` and especially `BalMCTS-GPT` highlight the robustness of our search-guided representation learning framework and its capacity to generalize across diverse and larger-scale CSPs.

For a more comprehensive evaluation of MIRROR's contribution to search node reduction, we present detailed ablation studies and supplementary results in Appendix D.2.

## 5 FINAL REMARKS

This paper presents `BalMCTS`, a Monte Carlo Tree Search (MCTS) framework tailored for training neural heuristics to optimize variable ordering in Constraint Optimization Problems (COPs). Our experiments on random COP instances show that `BalMCTS` reliably discovers high-quality solutions among the top 5 feasible candidates. Further evaluation on random Constraint Satisfaction Problems (CSPs) confirms that our approach achieves strong search efficiency, requiring substantially fewer node explorations to find solutions compared to classical heuristics. A key limitation, however, is the increased computational overhead during inference, which can offset the gains in node efficiency. Future work could explore leveraging domain knowledge—such as planning language descriptions—to further prune the search space and improve overall computational performance.

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

## A    INTERACTIVE VARIABLE SELECTION WITH LARGE LANGUAGE MODELS

**Prompt:**

Given a constraint optimization problem with the following parameters:

- Number of variables
- Domain length of each variable
- Number of constraints on each variable
- The mean value that can be taken by each variable
- Partial assignment

it is recommended to choose the next variable to be assigned. I'd like you to use your intuition to recommend a variable rather than existing heuristic algorithms, which could lead to the minimum search space and value.'

**Model Input:**

- Number of variables: 3
- Domain length of each variable: [2, 2, 3]
- Number of constraints on each variable: [3, 2, 3]
- The mean value that can be taken by each variable: [3, 4, 3]
- Partial assignment: []

**Model Output:**

Let's analyze the variables based on their given attributes:

1.**Variable 1:** - Domain length: 2 - Mean value: 3 - Number of constraints: 3

2.**Variable 2:** - Domain length: 2 - Mean value: 4 - Number of constraints: 2

3.**Variable 3:** - Domain length: 3 - Mean value: 3 - Number of constraints: 3

In constraint optimization problems, it is often useful to consider variables with the most constraints first because they have a higher impact on the possible solutions. This reduces the search space more effectively. Additionally, variables with smaller domain lengths often help in pruning the search space faster.

**Analysis**:

**Constraints:** Variables 1 and 3 have the most constraints (3). **Domain Length:** Variable 3 has a larger domain length (3) compared to Variables 1 and 2 (both with domain length 2).

Given this, variable 1 has the most constraints and a smaller domain length, which increases the chances of quickly reducing the search space through pruning.

Therefore, based on the number of constraints and the relatively smaller domain: **I recommend choosing Variable 1 as the next variable to be assigned.**

This choice balances between the constraints and the domain size, potentially leading to more effective search space minimization.

## B    SOLVING ALGORITHM

The solving algorithm, presented in Algorithm 2, is designed for solving COPs using a given $(X, D, C, A)$. Where $A$ represents assignments of variables. Its output is the objective value $\tau$. Lines 1-3 specify the termination conditions for the recursive. Lines 4 - 9 describe the process of determining the next variable to select by `BalMCTS` and `GPT`, where $\epsilon$ = (The number of constraints - the number of domains) / The number of constraints. Because Based on the suggestions provided by GPT, we can summarize the two patterns as follows:

1. Prefer variables with a smaller domain of definition.

---

**Algorithm 2** Solving

---

**Input**: $X, D, C, A$
**Output**: Objective $\tau$

1: **if** A is a feasible solution. **then**
2:     **return** $\tau$
3: **end if**
4: observe state $s$
5: **if** $\epsilon \geq \iota$ and $|A| \leq 3$ **then**
6:     $x = \text{GPT}(s)$
7: **else**
8:     $x = \text{BalMCTS}(s)$
9: **end if**
10: **for** $v \in sort(D(x))$ **do**
11:     $A \leftarrow A \cup \{(x = v)\}$
12:     **if** $\text{Solving}(X, D, C, A)$ **then**
13:         **return** $\tau$
14:     **end if**
15:     $A \leftarrow A - \{(x = v)\}$
16: **end for**

---

    2. Prefer nodes with more constraints.

Lines 10-15 involve assigning values to the variables. The sorting is based on the Least Constraining Value (LCV) and the extreme values of the neighborhood.

## C   THEOREM PROOF

### C.1   PROOF OF THEOREM 1

Define $P_{\text{root}}$ as the probability of finding a solution from the root node, where the search space includes all possible combinations:

$$P_{\text{root}} = \frac{|S|}{d^n}, \tag{8}$$

where $d^n$ are possible assignments and $|S|$ are solutions. At the current node, after fixing $m$ variables, the search space reduces to $d^{n-m}$ combinations, so the probability of finding a solution is:

$$P_{\text{current}} = \frac{|S_{\text{fixed}}|}{d^{n-m}} \tag{9}$$

To compare these probabilities, compute the ratio:

$$\frac{P_{\text{current}}}{P_{\text{root}}} = \frac{\frac{|S_{\text{fixed}}|}{d^{n-m}}}{\frac{|S|}{d^n}} = \frac{|S_{\text{fixed}}| \cdot d^n}{|S| \cdot d^{n-m}} = \frac{|S_{\text{fixed}}|}{|S|} \cdot d^m \tag{10}$$

For $P_{\text{current}} > P_{\text{root}}$, we require:

$$\frac{P_{\text{current}}}{P_{\text{root}}} > 1 \implies \frac{|S_{\text{fixed}}|}{|S|} \cdot d^m > 1 \tag{11}$$

Since $|S_{\text{fixed}}| \geq 1$, the minimum value of the left-hand side occurs when $|S_{\text{fixed}}| = 1$. Substituting this minimum value, we obtain:

$$\frac{1}{|S|} \cdot d^m > 1 \implies |S| < d^m \tag{12}$$

Thus, when $|S_{\text{fixed}}| \geq 1$ and $|S| < d^m$, the solution probability exceeds the root probability $P_{\text{current}} > P_{\text{root}}$. This is effective in sparse constraint optimization problems, where fixing variables via constraint propagation boosts local solution density.

## C.2   Proof of Theorem 2

The generalization error is define as:

$$\epsilon_g = |\mathbb{E}_{x,y\sim p(x,y)}[\mathcal{L}(f_\theta(x),y)] - \frac{1}{N}\sum_{i=1}^{N}\mathcal{L}(x_i),y_i)|, \tag{13}$$

where $\mathcal{L} : [0,1]^n \times [0,1]^n \to \mathbb{R}$ is the loss function. Since $y_i, f_\theta(x_i) \in [0,1]^n$, consider the mean squared error:

$$\mathcal{L}(f_\theta(x),y) = \frac{1}{n}\sum_{j=1}^{n}(f_\theta(x_j),y_j)^2. \tag{14}$$

As $|f_\theta(x_j) - y_j| \le 1$, we have $\mathcal{L} \in [0,1]$. Since $(x, y_i)$ from $p(x,y)$, apply Hoeffding's inequality Hoeffding (1994):

$$P(|\frac{1}{N}\sum_{i=1}^{N}\mathcal{L}(f_\theta(x_i),y_i) - \mathbb{E}[\mathcal{L}]| \ge \epsilon)$$

$$\le 2\exp\left(-2N\epsilon^2\right) \to \epsilon_g \le 2\exp\left(-2N\epsilon^2\right). \tag{15}$$

Thus, increasing the sample size N achieves low generalization error.

# D   Experiment

## D.1   Experimental Setup

**Instance generation.**   In our experiments, we consider three standard benchmark types: random Constraint Optimization Problems (COPs), Weighted Graph Coloring Problems (WGCP), and random Constraint Satisfaction Problems (CSPs). To randomly generate tasks, we employ the RB model Xu et al. (2007), utilizing five parameters $\langle m, n, \alpha, \rho, \beta \rangle$. It is noteworthy that, for generating CSP instances, an additional parameter $\Delta$ is introduced, and the generation process involves assigning a random cost from $[0, \Delta]$ to each pair of constraints. The specific purposes of these parameters are detailed below:

- $m \ge 2$ is the arity of each constraint;

- $n \ge 2$ is the number of variables;

- $\gamma > 0$ specifies $d$, which is the domain size of each variable, and $d = n^\alpha$;

- $\beta > 0$ specifies $e$, which is the number of constraints, and $e = \beta \cdot n \cdot \ln n$;

- $\rho \in (0,1)$ specifies the constraint tightness and $\rho \cdot d^k$ is the number of disallowed tuples for each constraint;

- $\Delta$ specifies the difference between the maximum and minimum values in the constraint.

Each unique combination of the above parameters delineates a class of CSP instances, which can be regarded as the distribution. The CSP instances employed are all positioned at the phase transition thresholds, where parameter combinations result in the most challenging instances. An advantageous theoretical property of the RB model, distinguishing it from other random CSP models, is its ability to ensure precise phase transitions and instance difficulty at the threshold Xu et al. (2007). We evaluate our approach across distribution $\langle 2, n, 0.7, 3, 0.21, \Delta \rangle$ for the random CSP and COP tasks, and on a distribution of $\langle 2, n, 0.7, 3, 0, \Delta \rangle$ for the WGCP task. Instances of varying scales are generated by adjusting the values of $n$ and $\Delta$ for different tasks. Specifically, in the case of random COP tasks, we opt for $n = \{20, 30, 40\}$ and $\Delta = 5$, indicating weights taking the values from $\{0, 1, 2, 3, 4, 5\}$. As for WGCP, we select $n = \{20, 30, 40\}$ and $\Delta = 1$, normalizing all weights to $[0, 1]$. For the random CSP, we choose $n = \{15, 20, 25, 30\}$ and $\Delta = 0$.

Table 3: Average gap of top-k solutions over different $n$.

| Random COPs $\langle m = 2, \gamma = 0.7, \beta = 3, \rho = 0.21, \Delta = 5 \rangle$ | | | |
|---|---|---|---|
| | n = 20 | n = 25 | n = 30 | n = 40 |
| k = 1 | 5.71% | 10.53% | 19.44% | 28.92% |
| k = 5 | 2.28% | 3.59% | 13.86% | 17.63% |
| k = 10 | 1.42% | 1.82% | 8.65% | 15.24% |
| k = 20 | 0.00% | 0.00% | 0.00% | 9.59% |

**Baselines.** For COPs baselines, we compare our proposed approach with the state-of-the-art COP solvers: **(1)** DABP with a splitting ratio of 0.95 Deng et al. (2022) **(2)** Toulbar2 with a timeout of 1200s Allouche et al. (2010) **3** Google OR-Tools. For CSPs baselines, we compare with the state-of-the-art CSP solvers such as DRL Song et al. (2022). Additionally, we evaluate against four classic hand-crafted variable ordering heuristics commonly employed in various CSP solvers: MinDom Haralick & Elliott (1980), Impact Refalo (2004) is default search strategy in OR-Tools, Dom/Ddeg Bessiere & Régin (1996) and Dom/Tdeg Li et al. (2016), which are utilized as features in our methods.

**Implementation details.** For our GNN model, we fix the embedding dimension at $p = 128$, and all MLPs have $L = 3$ layers with hidden dimension $64$ and rectified linear units (RELU) as activation functions. The embeddings are updated for $K = 5$ iterations. Our model is implemented with the PyTorch Geometric framework Fey & Lenssen (2019) and trained with the Adam optimizer Kinga et al. (2015) using the learning rate of $5 \times 10^{-5}$ and mini-batch size $\mathcal{B} = 128$. The size of experience replay is $\mathcal{M} = 1M$. The frequency of updating the target network is $e_u = 10$. We employ $\mathcal{P} = 100$ instances, and the simulation $\mathcal{T}_{\max}$ is limited to $10000$ with $\mathcal{N} = 10$, indicating that each expanded node will be simulated 10 times. For testing, we impose a cutoff limit of $5 \times 10^5$ search nodes for our policies. The implementation of our approach is carried out in Python. All experiments are conducted on an Intel(R)-E5-2637 with RTX 1080 GPU (11GB memory) and 251GB memory.

D.2 ADDITIONAL ANALYSES.

**Average gap of top-$k$ solutions.** In Table 3, we show the optimal gap that can be achieved for the top-$k$ solutions for the random COPs task, showing more clearly the efficiency of our model in finding high-quality feasible solutions that may be close to the optimal or best solution. It is evident that our method achieves an average gap of less than 30% in the first obtained solution, as previously indicated in Table 1, showcasing the significantly reduced number of nodes required compared to the Toulbar2 solver. This performance is noteworthy, with an average gap of less than 17.63% in the top 5 solutions and the ability to find optimal solutions in simple instances within the top 20 solutions. Our training objective is balancing objective function and search nodes, enabling our algorithm to quickly identify a near-optimal solution. This initial solution serves as a foundation for further branching to find the optimal solution, unlike other methods that prioritize branching before optimizing.

**Generalization Analysis of different herustic function Across Problem Scales** Figure 5 shows the proportion of instances where each method achieves the minimum number of search nodes across problem scales ($n = 15, 25, 30, 40$). Key observations include: (1). BalMCTS consistently achieves the minimum search nodes across all scales, especially in larger problems ($n = 30, 40$). Its combination of Monte Carlo Tree Search (MCTS) and heuristic strategies makes it highly adaptable and efficient. (2) While DRL performs well (ranking third in Tables 2, it has the smallest proportion in the figure. This is because BalMCTS, an improved version of DRL, outperforms it in most instances where DRL would excel, reducing DRL's standalone contributions. (3). Traditional heuristic methods (e.g., Dom/Tdeg, Dom/Ddeg) perform well in small-scale problems ($n = 15, 25$) but struggle as problem size increases. Their reliance on specific problem structures makes them less effective in larger, more complex instances. This experiment demonstrates the generalization capability and effectiveness of our method in solving Constraint Satisfaction Problems across varying scales and complexities.

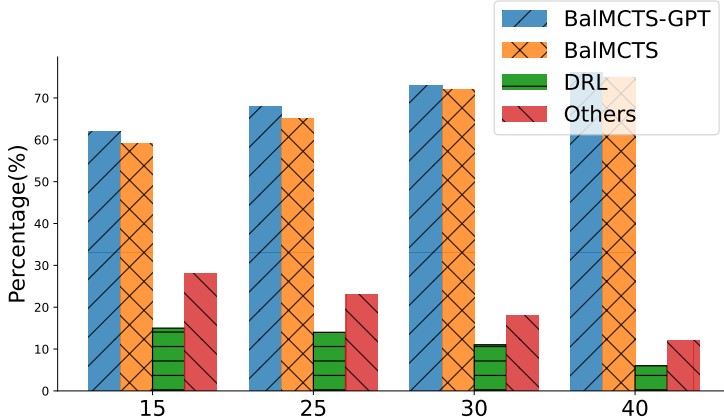

Figure 5: Proportion of Instances with Minimum Search Nodes Across Problem Scales. The figure shows the proportion of instances where each method achieves the minimum search nodes for problem sizes n=15,25,30,40. If multiple methods achieve the same minimum, the instance is counted for each method, providing a clear comparison of efficiency across scales. The "Others" category includes the Tdeg, Ddeg, MinDom, and Impact methods.

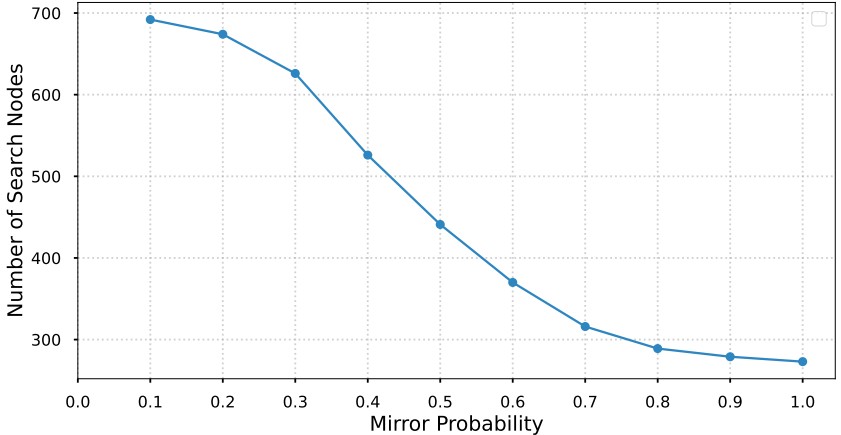

Figure 6: Impact of Mirror on search nodes with parameters $\langle m = 2, \gamma = 0.7, \beta = 3, \rho = 0.21, \Delta = 0, n = 25 \rangle$.

**Analysis of the MIRROR Operation:** To evaluate the impact of the MIRROR operation, we conducted ten experiments on model RB instances $\langle m = 2, \gamma = 0.7, \beta = 3, \rho = 0.21, \Delta = 0, n = 25 \rangle$, varying the probability of applying MIRROR after finding a solution. The x-axis of Figure 6 represents the probability of applying the MIRROR operation when a solution is found during the simulation phase, while the y-axis indicates the number of search nodes. The results show a consistent decrease in search nodes as the MIRROR probability increases, driven by an increase in the number of discovered solutions and the subsequent expansion of the training dataset. However, the MIRROR operation also increases the number of search iterations, extending training time.

