# OpenReview forum: "Balancing Objective Function and Search Nodes in MCTS  for Constraint Optimization Problems"
_ICLR.cc/2026/Conference — Submitted to ICLR 2026_

### Official Review · Reviewer_Fp92 · 2025-10-16

**Soundness:** 2
**Presentation:** 1
**Contribution:** 2
**Rating:** 2
**Confidence:** 3

**Summary:**

In this paper, the authors approach Constraint Optimization Problems, and do it by proposing an MCTS model that uses a GNN for guiding better while selecting variables. The authors address a historical problem faced by algorithms that use search trees to solve problems. Depending on which variables are chosen to set values, the path to finding a feasible solution can be greatly extended. In this sense, finding the shortest path is a real challenge.

**Strengths:**

S1. The problem addressed is relevant.

**Weaknesses:**

W1. The authors place great emphasis on the idea of reducing the number of nodes explored during the search and the Node Efficiency (NE) measure. However, when one reads the results in Table 1 and sees that OR-tools obtains optimal solutions in less time than BalMCTS... I wonder what the point of reviewing more or fewer nodes is if, in the end, a classic and infinitely simpler solver solves it faster. I'm not saying that the work proposed in this paper isn't interesting, but the overhead introduced by BalMCTS clearly doesn't compensate for it.

W2. The authors do not use the references correctly. Example: In line 032, it should read (Modi et al. 2025) instead of Modi et al. (2025). The same applies to line 033 and many other places in the article. By the way, why is the citation in line 86 switched to another one?

W3. Typos in lines 074 and 075.

W4. It is not explained/well-motivated why the three principles introduced in the paragraph above Fig. 1 are relevant. In fact, it is not clear how the contributions that appear afterward are related to them.

W5. The paper should be carefully revised and the writing improved. There are strange statements across the paper. For example, in line 080, the authors explain their interest in finding a universal solver. This is the holy grail of optimization. But it has long been known that it clashes head-on with the No-Free-Lunch theorem (Wolpert and Macready, 1997). In addition, I would not say that COPs are, by nature, dynamic.

W6. Why italics in lines 122? The paper has many non-justified formatting issues. It looks like the manuscript is still to be polished.

W7. In the experimental section, the authors have moved a lot of material to appendices, making the main body confusing in some parts. For example, what are the Delta symbols in the caption of Table 1? At which moment did BalMCTS-GPT appear? What is WGCP? Note that the main body should be self-contained and understandable. Then, if the reader aims to know more details, he/she can read the appendices.

W8. Why is OR-Tools not included in Table 2?

**Questions:**

In addition to those in weaknesses:

Q1. Did the authors try to optimize larger instances?

Q2. Why does OR-tools explore many more nodes and take less time than BalMCTS?

---

### Official Review · Reviewer_kFpe · 2025-10-26

**Soundness:** 2
**Presentation:** 2
**Contribution:** 2
**Rating:** 2
**Confidence:** 3

**Summary:**

This paper proposes BalMCTS, a framework that integrates reinforcement learning with Monte Carlo Tree Search (MCTS) to solve Constraint Optimization Problems (COPs) more efficiently. The key idea is to use a learned Q-value function, trained via a double DQN setup, to guide node selection within the MCTS process. By leveraging graph neural network (GNN) embeddings of constraint graphs and a data-augmentation strategy called MIRROR, the method aims to balance solution quality with search efficiency. Experimental results on random COP and CSP benchmarks indicate that BalMCTS reduces the number of explored nodes compared to established solvers such as Toulbar2, OR-Tools, and DABP.

**Strengths:**

1. **Clear motivation and problem setting**
   The paper addresses a well-defined goal of improving search efficiency for Constraint Optimization Problems (COPs) by integrating learning-based heuristics with Monte Carlo Tree Search (MCTS).

2. **Integration of GNN and MCTS**
   The idea of embedding COP structures using Graph Neural Networks and coupling them with MCTS is conceptually interesting.

3. **Inclusion of MIRROR data augmentation**
   The proposed MIRROR operation for generating diverse variable orderings is a reasonable attempt to mitigate sparse reward issues in combinatorial domains.

**Weaknesses:**

**Weakness 1 — Limited impact of the proposed Q-value guidance**

The learned Q-value in BalMCTS seems to play only a minor role in improving the overall search process. The network is used solely to determine the **order of variable selection** within MCTS, while the **value assignments are chosen randomly**. Consequently, the policy affects only the sequencing of variable instantiation, not the actual decision that influences the objective value. This considerably limits the contribution of learning. It would be helpful if the paper explicitly clarified this point to avoid confusion among readers regarding what aspect of the search process is actually being guided.

From a conceptual standpoint, variable ordering alone is not a decisive factor in solving COPs—given any fixed order, an optimal solution can still be found through exhaustive search. Although a better order may reduce the number of visited nodes to some extent, it does not fundamentally affect solution quality or convergence. In contrast, learning to **select effective value assignments** for each variable would represent a much stronger form of neural guidance. Overall, the proposed idea provides only a limited improvement to the effectiveness of MCTS in COPs.

---

**Weakness 2 — Incomplete and insufficient feature representation**

The proposed GNN encoder fails to capture sufficient information about the underlying constraint structure of the COP. Each constraint node is represented by only three scalar features—number of bound variables, dynamic tightness, and minimum cost—which provide an overly coarse description of the constraint.

1. The *minimum cost* value does not reveal *which variable assignments* achieve that cost.
2. It also ignores the *range and distribution of costs* for other possible assignments within the same constraint.

Due to this limited representation, the GNN lacks visibility into the true landscape of each constraint and cannot accurately identify promising variable–value pairs. The model therefore relies on incomplete signals and is unlikely to learn optimal Q-values. In short, I believe that the feature design is too simplistic to support meaningful decision-making within the proposed framework.

**Questions:**

1. **Training dynamics and convergence**
   During training, did you observe a clear convergence of the loss value or Q-value estimates?
   How many distinct COP instances were used for training, and how sensitive is performance to the training set size?
   A plot of the training curve (e.g., loss or reward versus iterations) would help clarify whether the model stabilizes or oscillates during learning.

---

2. **Runtime and computation flow**
   For evaluation, my understanding is that the Q-value must be evaluated by the neural network for each node expansion within the MCTS. Is this correct?
   If so, it is surprising that the reported BalMCTS runtime is relatively short. The MCTS traversal logic runs on the CPU while the Q-value computation is on the GPU, so data transfer between the two could introduce nontrivial latency. Could you clarify how this interaction is implemented and whether asynchronous batching or other optimizations were used to keep the runtime low?

---

### Official Review · Reviewer_QnzL · 2025-10-28

**Soundness:** 2
**Presentation:** 1
**Contribution:** 2
**Rating:** 2
**Confidence:** 3

**Summary:**

The paper suggests BalMCTS, which integrates a GNN-based heuristic into Monte Carlo Tree Search to learn variable ordering for COPs/CSPs. targeting a balance between solution quality and the number of explored nodes.
It trains online with a DDQN target and uses a MIRROR augmentation that reorders variables of found solutions to generate more diverse training paths within the MCTS loop.
On RB-generated random COPs, WGCP, and CSPs, BalMCTS (and a BalMCTS-GPT variant) generally achieves fewer search nodes than classical heuristics.

**Strengths:**

* Integrating learnable components into variable selection heuristics remains an important area of study.
* The GNN consumes problem-agnostic signals (domain size, assigned flag; constraint tightness, etc.), which should in principle transfer across COP families without heavy feature engineering.
* The ablation in the Appendix supports that the number of search nodes reduces as the MIRROR probability increases, validating its effectiveness as an augmentation strategy.

**Weaknesses:**

* Section 3.4 suggests using LLMs to select variables in the search process. The paper lacks a clear motivation for why this would be desirable. In general, the latency of prompting an LLM should far exceed the cost of a GNN call or a classical variable selection heuristic. Based on Appendix B the LLM seems to make decisions based on two simple heuristics using the domain sizes and the number of constraints. Why is it necessary to prompt an LLM to follow such simple heuristics? Would it not be a better use of frontier LLMs to let a coding agent integrate these heuristics into the source code of existing solvers to avoid the cost and noise of re-prompting LLMs during the search process? The paper also lacks details on how exactly GPT is integrated, i.e. the the $\iota$ parameter is never defined.
* The Theory in Section 3.5 is too generic and lacks details. Theorem 1 does not specify formally what is meant by the "probability of finding a solution" from a given node. Any complete search algorithm will always find a solution if one exists, so it is not clear from the theorem's statement what is meant here. Theorem 2 is a generic Hoeffding bound not tailored to the model or data generation process. Neither really explains why MIRROR or the specific loss should work well.
* The scope of the experimental evaluation is very narrow. All results are on random COP/CSP and WGCP instances with uniform random structure. The performance on more structured problems is not evaluated.
* The paper uses the number of search nodes as the main cost measure even though wall-clock runtime is ultimately most important in practice. Providing data on the objective value of COPs as a function of the wall-clock runtime would be more insightful than the "time to first feasible solution" of Table 1.
* The metric used in Table 2 is not specified. Is this the number of search nodes, the wall-clock runtime, or some other cost measure?
* The paper contains many typos (“priciples”, “adative”, “envionment”, “herustic”, “Kinga, Jimmy Ba Adam”).
* Key hyperparameters (c1 in UCT; c3,c4 in y, $\iota$) are not specified.

**Questions:**

* What is the motivation for directly prompting an LLM to make variable selection decisions?
* What exactly is the "probability of finding a solution" referring to in Theorem 1?
* How does the objective function of COPs change as a function of wall-clock runtime for the compared methods?
* What metric is used in Table 2?

---

### Meta-Review · Area_Chair_Nbf9 · 2025-12-29

**Summary:**

This work proposes BalMCTS, a hybrid framework that integrates Monte Carlo Tree Search (MCTS) with a GNN to efficiently solve constraint optimization problems by learning adaptive variable-ordering heuristics. It further introduces the MIRROR operation for data augmentation, which can significantly reduce the search nodes and improve generalization, especially in sparse and large-scale problem instances.

The reviewers have consistent negative scores (2,2,2) on this work and raised many concerns on motivation, impact of the proposed method, feature representation,  LLM-based variable selection, training dynamics, theory, the scope and settings of experiments, clarifications, reference format and typos. However, the authors did not provide a rebuttal to address any of these concerns.

Therefore, I recommend rejecting this work.

**Reviewer Concerns:**

None of the reviewer concerns have been properly addressed since no rebuttal has been provided.

**Reviewer Scores:**

I believe all reviewers will keep their original negative score (2,2,2) due to the missing rebuttal.

---

### Decision · Program_Chairs · 2026-01-26

Reject